# Clathrate Hydrates of Organic Solvents as Auxiliary Intermediates in Pharmaceutical Research and Development: Improving Dissolution Behaviour of a New Anti-Tuberculosis Drug, Perchlozon

**DOI:** 10.3390/pharmaceutics14030495

**Published:** 2022-02-24

**Authors:** Andrey G. Ogienko, Svetlana A. Myz, Andrey A. Nefedov, Anna A. Ogienko, Tatyana P. Adamova, Olga M. Voronkova, Svetlana V. Amosova, Boris A. Trofimov, Vladimir V. Boldyrev, Elena V. Boldyreva

**Affiliations:** 1Faculty of Natural Sciences, Novosibirsk State University, 630090 Novosibirsk, Russia; anefyodov@yandex.ru (A.A.N.); v.boldyrev1@g.nsu.ru (V.V.B.); 2LLC “SP IIC”, 630049 Novosibirsk, Russia; myz.svetlana@pm-hm.ru (S.A.M.); helga98.95@mail.ru (O.M.V.); 3N.N. Vorozhtsov Novosibirsk Institute of Organic Chemistry, Siberian Branch of the Russian Academy of Sciences, 630090 Novosibirsk, Russia; 4Institute of Molecular and Cellular Biology, Siberian Branch of the Russian Academy of Sciences, 630090 Novosibirsk, Russia; ogienko.anna@gmail.com; 5A.V. Nikolaev Institute of Inorganic Chemistry, Siberian Branch of the Russian Academy of Sciences, 630090 Novosibirsk, Russia; adamova@niic.nsc.ru; 6A.E. Favorsky Irkutsk Institute of Chemistry, Siberian Branch of the Russian Academy of Sciences, 664033 Irkutsk, Russia; amosova@irioch.irk.ru (S.V.A.); boris_trofimov@irioch.irk.ru (B.A.T.); 7V.V. Voevodsky Institute of Chemical Kinetics and Combustion, Siberian Branch of the Russian Academy of Sciences, 630090 Novosibirsk, Russia; 8G.K. Boreskov Institute of Catalysis, Siberian Branch of the Russian Academy of Sciences, 630090 Novosibirsk, Russia

**Keywords:** formulations for inhalation, formulations for injections, freeze-drying, anti-tuberculosis drugs, clathrate hydrates

## Abstract

There is an urgent need for new drugs to overcome the challenge of the ever-growing drug resistance towards tuberculosis. A new, highly efficient anti-tuberculosis drug, Perchlozone (thioureidoiminomethylpyridinium perchlorate, Pz), is only available in an oral dosage form, though injectable forms and inhalation solutions could be better alternatives, offering higher bioavailability. To produce such forms, nano- and micro-particles of APIs would need to be prepared as dispersions with carriers. We use this case study to illustrate the principles of selecting solvents and excipients when preparing such formulations. We justify the choice of water–THF (19.1 wt % THF) as solvent and mannitol as carrier to prepare formulations of Pz—a poorly soluble compound—that are suitable for injection or inhalation. The formulations could be prepared by conventional freeze-drying in vials, making the proposed method suitable for industrial scaling. A similar strategy for selecting the organic solvent and the excipient can be applied to other compounds with low water solubility.

## 1. Introduction

Tuberculosis (TB), caused by *Mycobacterium tuberculosis*, remains a major threat to global health. In 2017 alone there were 6.6 million new cases of TB, including 600,000 new cases of multidrug-resistant (MDR) or rifampicin-resistant (RR) TB [1]. The COVID-19 pandemic has reversed years of global progress in tackling tuberculosis and for the first time in over a decade TB deaths have increased, according to the World Health Organization’s 2021 Global TB report. Approximately, 1.5 million people died from TB in 2020 [2].

The current anti-TB drug development pipeline is limited in terms of new chemical entities. Most compounds that reach clinical trials are either repurposed drugs, or analogues of existing drugs [3]. There is an urgent need for new drugs with new mechanisms of action to overcome the challenge of the ever-growing drug resistance towards TB.

There are two main options for developing new medications: (1) to design and develop new active pharmaceutical ingredients (APIs) and (2) to design and develop new formulations with a higher bioavailability. The latter can, for example, be achieved by preparing formulations suitable for new methods of administration. Importantly, the two directions are not mutually exclusive. On the contrary, an optimum formulation needs to be developed for each new API. An obvious recent trend in drug formulation development is to make them suitable for inhalation or intranasal delivery [4,5,6,7,8,9,10,11,12,13,14,15,16,17]. One of the major problems of treating pulmonary diseases is that antibiotics administered perorally, or even via injections, cannot easily penetrate the lungs; their therapeutic concentration cannot therefore be achieved in the pulmonary tissues. This results in an increase in the resistance of the relevant microorganisms to antibiotics and also increases hepato- and nephrotoxicity [18,19].

A new highly efficient anti-tuberculosis drug, Perchlozone (thioureidoiminomethylpyridinium perchlorate, Pz), has been developed at the A.E. Favorsky Irkutsk Institute of Chemistry SB RAS and the Saint-Petersburg Research Institute of Phthisiopulmonology (Figure 1) [20]. Pz is active against MDR. It has successfully passed clinical studies and is approved [21] for medical applications. In 2013, industrial production of Perchlozone based on the method described in the patent [22] has been launched at JSC Pharmasyntez (Moscow, Russia) [23]. Additionally, a more environmentally friendly method for the synthesis of the Perchlozone pharmacopoeial drug [24] has been developed. The method allows the obtainment of a cleaner substance (99%) in aqueous medium at room temperature. Recent research confirms the high efficiency of this drug [25,26,27,28,29,30].

Pz is currently administered in an oral dosage form (tablets), though injectable forms and inhalation powders could offer higher bioavailability and possibly lower hepatotoxicity [27,28,29,30]. To produce such forms, nano- and micro-particles of APIs would need to be prepared as dispersions with carriers. This helps to increase the bioavailability of poorly soluble and poorly wettable APIs by improving their dissolution profile and solubility. Furthermore, these particles help to overcome the difficulties associated with static charging, inadequate flow properties, and particle aggregation [31,32,33,34,35]. Solid dispersions can be prepared by various techniques, including recrystallization, mechanical milling, spray drying, supercritical fluid processing, and freeze-drying. Freeze-drying is often the technology of choice when processing compounds that decompose upon heating or with mechanical treatment. In many cases, this technique may be less damaging with respect to decomposition than other micronization techniques. It is also useful in many cases to achieve control over the polymorphism. Polymorphs obtained by freeze-drying may differ from those formed by spray-drying or with mechanical treatment. For example, spray drying of glycine aqueous solutions gives the α-polymorph, mechanical treatment yields a mixture of the α- and γ-polymorphs, whereas freeze-drying leads to the formation of the β-polymorph. [36]. Unfortunately, many promising compounds are hydrophobic, and the negligible aqueous solubility of these compounds greatly limits the application of freeze-drying. In such cases, water must be substituted for organic solvents, leading to higher financial costs and potential for chemical hazards during manufacturing and drug administration. Unfortunately, the melting points of almost all Class 3 residual solvents (solvents that do not cause a human health hazard at levels normally accepted in pharmaceuticals) are too low (below −70 °C; see details in Appendix A) to make these solvents suitable for freeze-drying [37,38,39].

During lyophilization the product temperature must remain below the characteristic glass transition/eutectic melting temperatures (T_g_’/T_E_) of amorphous/crystalline phases. To reduce processing costs, the primary drying (sublimation) stage must occur at as high a temperature as possible. The sublimation temperature depends critically on the formulation details [40,41,42]. To allow for a reasonable drying time, the T_g_’/T_E_ should be no lower than −40 °C [43].

An attractive alternative for using pure water is substituting water for a water–organic mixture, in which the API is more soluble. Freeze-drying from such a mixed solvent can be cheaper and yield safe high-quality pharmaceutical products [37,39]. Many water–organic solutions form clathrate hydrates on freezing [44]. Clathrate hydrates are water-based crystalline compounds in which water molecules form a hydrogen-bonded tetrahedral framework as a space-filling packing of polyhedra with trivalent vertices (simple polyhedra). The vertices of the polyhedra correspond to water oxygen atoms and the edges are hydrogen bonds. Within the cavities of the host lattice, small (0.35 ÷ 0.9 nm) non-polar guest molecules of gases or volatile organics are incorporated. The host–guest interactions in this case are purely van der Waals [45,46]. Many of these clathrate hydrates are stable up to relatively high temperatures. For example, the clathrate hydrate of tetrahydrofuran (THF) is stable up to temperatures of about +4.5 °C, while the eutectic “THF hydrate–THF” melting point in this system is −109 °C [44]. In our earlier publications, we described the technique of obtaining fine particles of both individual compounds and their solid dispersions using freeze-drying. In this method, API solutions are freeze-dried in organic–water systems. Freezing gives organic–water clathrate hydrates, with the API remaining finely dispersed outside the clathrates. In the primary drying stage the clathrates sublime, leaving the pure API in a finely dispersed form, if no liquid phases are formed on heating which could cause a recrystallization into larger particles [47,48,49,50]. The main difficulties with this technique are the selection of an organic solvent, the optimization of the water–organic solvent ratio and the identification of a temperature–pressure protocol that avoids the formation of any liquid phases on heating. This cannot be easily done by trial-and-error. Instead, this optimization procedure requires detailed thermodynamic and structural investigations [48,49,50].

Herein we report a method for producing Pz formulations with improved dissolution behavior, suitable for intravenous or respiratory (via nebulizers) delivery. Production is based on freeze-drying solutions of Pz with pharmaceutically acceptable excipients in a mixture of water with the organic solvent THF.

## 2. Materials and Methods

### 2.1. Materials & Samples Preparation

Pz was synthesized according to [22]. It was obtained as the only reported crystalline phase documented in the Cambridge Structural Database as RUKGUA [51]. d-(+)-trehalose dihydrate (for microbiology, ≥99.0%; Sigma-Aldrich, cat. no. BCBW5088) (*P*2_1_2_1_2_1_, [52]) and mannitol (analytical grade, JSC “Reakhim”, *P*2_1_2_1_2_1_, [53]) were used as purchased (β-polymorph). Glycine (analytical grade, JSC “Reakhim”) (mixture of α- and γ-polymorphs) was recrystallized from aqueous solutions as described in [54] to prepare the pure α form (space group *P*2_1_/*n*). Peroxide-free twice-distilled THF (JSC “VEKTON”) and distilled water were used as solvents.

### 2.2. Methods

#### 2.2.1. Freeze-Drying

Freeze-drying was accomplished with a laboratory-scale freeze-dryer (NIIC SB RAS, Novosibirsk, Russia). Convection-enhanced Pirani gauges (275 Mini-Convectron^®^ (Granville-Phillips^®^, MKS Instruments, Inc., Longmont, CO, USA)) were used to monitor the chamber and condenser pressure. The shelf was equilibrated at −10 °C before the vials or drying trays were loaded and then placed under vacuum. Freeze-drying was carried out at a chamber pressure of 100/400 mTorr (dry nitrogen) for ~10 h. The shelf temperature was then increased to 30 °C and held at this temperature for 2 h. The pressure in the freeze-dryer was subsequently increased to *p* = 1 bar (ambient) by filling it with dry nitrogen.

##### Ultrafine Pz Preparation Method (Thin Film Freezing (TFF) + Freeze-Drying (FD))

Pz (300 mg) was dissolved by stirring at +30 °C in 20.0 g of the THF–water mixture (19.1 wt % of THF, THF hydrate composition) in a transparent borosilicate glass vial (Sci/Spec, B75525). The hot solution was splashed as a thin layer onto a copper plate cooled to liquid nitrogen temperatures. After that it was ground in a mortar and placed onto a drying tray cooled to liquid nitrogen temperatures. The tray was then placed on a shelf that was pre-cooled to −10 °C and freeze-dried. Upon removal from the freeze dryer, samples were stored in a glass vial closed tightly with a tetrafluorethylene (TFE) cap.

#### 2.2.2. Pz Formulations Preparation Method

Pz (320 mg) and α-glycine–mannitol–trehalose dihydrate (320/640/1280 mg) were dissolved on stirring at +30 °C in 40.0 g of the THF–water mixture (19.1 wt % of THF). The obtained solution was immediately put into:(a)Preliminary experiments: 10 vials (1.00 mL aliquots; 5 mL vials, Sci/Spec, B69308);(b)Main experiments: 20 vials (2.00 mL aliquots; 15 mL injection vials made of colorless borosilicate glass tubing with 13 mm 2 leg freeze-drying stoppers (Jiangsu Runde Ltd., Jiangsu, China)) and frozen in air thermostat at −20 °C. Vials were placed on a shelf, which was pre-cooled to −10 °C, and freeze-dried at a chamber pressure of 100/400 mTorr.

#### 2.2.3. Thermoanalytical (TA) Experiments

The TA experiments were carried out using an instrument described in [50] (Appendix A). The signals were registered using a precise converter of signals of resistance thermometers and thermocouples “TERCON” with a switch of input signals “TERCON-K” (TERMEX, Tomsk, Russia). The set of samples was placed in the autoclave. Then the autoclave was cooled/heated (+25 °C → –20 °C (exposure for 30 min) → +25 °C) at a constant rate of 0.5 °C/min at atmospheric pressure. The absolute temperature uncertainty was ± 0.2 °C. Each of the two aluminum sample holders was loaded with four PTFE cells filled with solutions (the volume of a samples was 1.000 mL; the thickness of the solution layer in each of the cells was 13 mm). A type K thermocouple was placed in each of the samples with the junction placed approximately in the middle of the sample.

Starting solutions: aqueous glycine–mannitol–trehalose solutions (5 wt %) (reference solutions); THF–water mixture (THF hydrate composition, 19.1 wt % of THF) (reference solution); glycine–mannitol–trehalose–Pz (5/5/5/1.5 wt %) in the THF–water mixture (19.6 wt % of THF).

#### 2.2.4. X-ray Powder Diffraction Analysis

PXRD experiments aimed to (1) identify the phases formed on cooling the glycine–mannitol–trehalose solution in the THF–water mixture and (2) characterize the final solid products. A Bruker D8 Advance diffractometer (λ = 1.5406 Å, tube voltage of 40 kV and tube current of 40 mA, https://www.bruker.com/ (accessed on 20 February 2022)) was used, equipped with an Anton Paar TTK 450 low temperature chamber, which permitted working under a vacuum down to 10^−3^ Torr. The vial with the frozen solution was broken, the sample was gently ground in a mortar (all operations being performed at liquid nitrogen temperature) and placed onto a holder, which had been preliminarily cooled to −100 °C.

#### 2.2.5. Residual THF Level Determination

Residual THF levels were determined by GC–MS. Samples were dissolved in DMSO before measurements. GC–MS analysis was performed using an Agilent 6890N gas chromatograph (https://dv-expert.org/laboratornoe-oborudovanie/agilent (accessed on 20 February 2022)) coupled with an Agilent 5973N mass detector and automatic sample injector (Agilent 7683 autosampler). The sample components were separated in a Restek Rxi-5ms non-polar capillary column (30 m long by 0.25 mm diameter, with film thickness 0.25 μm of (5%-phenyl)-methylpolysiloxane) under the following conditions: probe volume 1 μL; split ratio 20:1 injection; 1 cm^3^/min helium flow rate (vacuum compensation); 280 °C GC injector temperature; 280 °C GC–MS transfer line temperature. The oven temperature was programmed between 50 °C (hold 2 min) and 280 °C at 10 °C/min (hold 3 min). The total ion current (TIC) chromatograms were obtained using an Agilent 5973N single quadrupole mass detector under the following conditions: electronic ionization at 70 eV; positive ion polarity; mass range *m*/*z* 12–100 amu; 230 °C MS ion source temperature; 150 °C quadrupole temperature. The GC–MS data were interpreted, and the peaks were identified using the Agilent MSD ChemStation software, version E.02.02.1431 (https://www.labx.com/product/chemstation (accessed on 20 February 2022)) and the NIST/EPA/NIH Mass Spectral Library (EI), data version NIST 11, https://www.sisweb.com/software/ms/nist.htm (accessed on 20 February 2022). The residual THF level was determined using an external standard.

#### 2.2.6. Scanning Electron Microscopy (SEM)

Morphological examination of the surface and internal structure of the samples was carried out using an EVO MA10 (Carl Zeiss) scanning electron microscope (https://www.zeiss.ru/corporate/ru/home.html (accessed on 20 February 2022)). The samples were mounted on a metal stub with double-sided adhesive tape and coated with gold to a thickness of about 8 nm using a Jeol JFC-1600 Auto Fine Coater.

#### 2.2.7. Elemental Analysis

Elemental analyses (C, H, N, S) were performed with a vario MICRO cube micro elemental analyzer (http://xn--80aajzhcnfck0a.xn--p1ai/PublicDocuments/1007076.pdf (accessed on 20 February 2022)) at the Analytical Laboratory of NIIC SB RAS (Novosibirsk). The total chlorine content was determined by the Sheniger method.

#### 2.2.8. Nebulizer Test

To find the maximum possible solution concentration that can be prepared when dissolving the lyophilizate of Pz with mannitol, we used either a saline solution (0.9 wt % of NaCl) or water for injections. The corresponding liquid (4 mL/2.67 mL/2 mL) was added to the lyophilizate (which contained 16 mg of Pz), yielding calculated Pz solution concentrations of 4 mg/mL, 6 mg/mL, and 8 mg/mL, respectively. To prepare the 4 mg/mL solutions, the dissolution was complete within 3–5 s of active shaking. In contrast, active shaking was insufficient for preparation of solutions with 6 mg/mL or 8 mg/mL, at least over the period of several months of observations. The stability of the supersaturated Pz solution with respect to membrane vibrations when used in nebulizers was tested. For this purpose, we used a mesh nebulizer (UN-233, A&D, Japan, http://www.and-rus.ru/ (accessed on 20 February 2022)); a filling volume ranging from 2 to 10 mL). Then, 4 mL of saline solution, or water for injections, were added to a vial that already contained a given Pz-mannitol formulation. After reconstitution (2–3 s), the solution was placed in the nebulizer reservoir and weighted. The mouthpiece was held stationary in a horizontal position for the duration of the test. After the operating period, the nebulizer reservoir was weighted again, and Pz concentrations in the remaining solutions were determined with HPLC fitted with a UV detector. The emitted dose (ED) was calculated from the ratio between the initial and final Pz content in the nebulizer reservoir.

#### 2.2.9. Determination of Pz Content

Pz content in the formulations was determined by HPLC. The HPLC analysis was performed using a Milikhrom A-02 device (ZAO EcoNova, Novosibirsk, Russia) equipped with a ProntoSIL 120-5C18 AQ analytical column (2.0 mm × 75 mm i.d., 5 μm particle size) (ZAO EcoNova, Novosibirsk, Russia). The mobile phase was composed of acetonitrile, 50 mM phosphate buffer with 5 mM octane sulphonic acid sodium salt, pH = 2.5, 15:85 of volume. The flow rate was 100 µL/min. The temperature of the column was maintained at 35 °C, and the effluent was monitored at 346 nm. A standard Pz solution was prepared by dissolving the weighted amount (5 mg) in methanol (25 mL) followed by storage in a refrigerator. To obtain a calibration curve, the volume was adjusted up to the mark with distilled water to give stock solutions. All measurements were performed in triplicate.

Figure 2 presents the outline of the experiments, explaining the general logic and justification for the selection of reactants and protocols.

## 3. Results and Discussion

### 3.1. The Solvent Choice 

THF was selected as a candidate for the mixed solvent system owing to its various beneficial properties over other Class 2 or 3 solvents. First, although Pz is highly soluble in DMF (a Class 2 residual solvent), DMF has a low melting point (−61 °C), rendering it unsuitable for freeze-drying. Similarly, Pz is well soluble in pure DMSO (a Class 3 residual solvent), although one must work with specially dried DMSO to avoid practical complications. For example, “wet” DMSO does not freeze completely in common laboratory refrigerators: the melting temperature of the frozen solution decreases significantly in the presence of water [55], and “DMSO–water” liquid inclusions remain within the frozen matrix, contaminating the final product with residual solvents. The usage of a mixed DMSO–water solvent is also excluded because of the low eutectic/peritectic temperatures in all the concentration ranges [55].

Pz has good solubility (>2 wt %) in THF–water mixtures (19.1 wt %, THF hydrate composition). Moreover, such THF mixtures form clathrate hydrates upon freezing with a melting point of +4.5 °C. This is an excellent temperature for efficient freeze-drying to obtain a solid dispersion of the Pz with excipients. Unfortunately, as a Class 2 solvent [37,38,39], residual THF levels in the final dispersed Pz product must be carefully controlled.

### 3.2. Trial Experiments with Precipitation of Pure Pz from Frozen THF–Water Solutions 

The fine Pz powder obtained by TFF + FD from a THF–water solution consisted of micrometer-sized “noodles” (linear dimensions: length >20 μm, width ca 400 nm; thickness ca. 100 nm) (Figure 1a,e and Appendix A). Pz precipitated as a crystalline phase, even on rapid freezing (Appendix A). The crystal structure of this phase was the same as that of the starting Pz powder [51]. The fact that the fine powder of Pz is crystalline is very important for practical applications. While amorphous drugs usually have higher solubility and higher dissolution rates than their crystalline counterparts [56,57,58,59], they remain metastable. Therefore, the solubility remains higher, only until the transformation into the thermodynamically more stable crystalline form inevitably occurs [56,60]. Spontaneous crystallization of the amorphous phase, as well as crystallization of more stable polymorphs from solution, are both very dangerous for pharmaceutical products [61,62]; therefore, whenever possible, amorphous phases are avoided.

### 3.3. Selection of a Carrier of Pz for the Two-Component Solid Dispersions 

The next step was to select a pharmaceutically acceptable carrier of Pz for a two-component solid dispersion. Preparing a solid dispersion is one of the most promising strategies to improve the oral bioavailability of poorly water-soluble drugs [63]. There are several reasons why such a dispersion is preferable as compared to the fine powder of pure Pz. First, Pz fine powder is prone to electrostatic charging, and is therefore very difficult to manipulate. Such powders have zero flowability and cannot be used to fill capsules for capsule DPIs (dry powder inhalers). Due to a very low powder density, the emitted dose will be extremely low (<5 mg even in the case of capsule size “00”). Second, because of its very poor aqueous solubility, the highest possible Pz concentration in a mixed THF–water solution is very low (1.5 wt %). In practice, this may result in Pz powder being blown from vials during primary drying, and hence in vial-to-vial variation in the deliverable dose and poor container-closure integrity.

Among small-molecule water-soluble excipients that are allowed for intravenous or respiratory (via nebulizers) delivery, we selected mannitol, trehalose, and glycine as potential candidates. They are often used in freeze-drying as bulking agents/carriers. They are highly soluble in the THF–water solution of the composition corresponding to the formation of the THF hydrate on freezing.

#### 3.3.1. Optimizing the Freeze-Drying Protocol Based on the TA and XRD Experiments 

The THF hydrate incongruent melting temperature depends on eutectic melting temperatures in the binary systems “third component–ice Ih” [48,50]. In this work we had four components in the system whose phase behaviour required explicit experimental study. The results of the TA experiments with the quaternary system “Pz–carrier–THF–water” are summarized in Table 1 for the carriers mannitol, trehalose, and glycine.

No thermal events were observed in the trehalose–THF–water system, confirming the absence of incongruent melting. This agrees well with the results of low-temperature PXRD experiments, which suggested that trehalose formed an amorphous freeze concentrate on cooling (Appendix A). The DSC profile of the frozen trehalose solution in THF—the water co-solvent system showed a glass transition with onset at ~−34 °C (Appendix A). The reflections of ice in the powder diffraction patterns of the frozen aqueous trehalose solution (Figure 2b) were absent not because the THF hydrate formation was complete, but because an amorphous freeze concentrate was formed.

Heating the sample of the frozen mannitol solution in THF–water from −100 to −10 °C does not change the intensities of the THF hydrate and ice Ih reflections (Figure 2a, diffraction patterns 2 and 3). Decreasing the pressure leads to the incongruent sublimation of the THF hydrate, which decomposes to ice Ih and gaseous THF (Figure 2a, diffraction patterns 3, 4, and 1b, diffraction pattern 4), as was recently shown in [64]. To achieve a maximum possible sublimation rate at a given sample temperature [65], the shelf temperature during the primary drying was set to −10 °C at 100 mTorr for the trehalose-based formulation and to −10 °C at 400 mTorr for the glycine- and mannitol-based formulations, based on data summarized in Table 1 and Figure 2.

Cooling glycine and mannitol solutions (5 wt %) in the THF–water co-solvent system (19.1 wt % of THF) in vials resulted in the crystallization of the metastable β-glycine (space group *P*2_1_ [66,67]) and mannitol hemihydrate, even though, originally, glycine was taken as the α-polymorph [68] and mannitol as the anhydrous β-polymorph [53], respectively. The THF hydrate formed almost quantitatively in the case of the glycine–THF–water system (Figure 2a, diffraction pattern 1). Crystallization of THF hydrate was partially inhibited in the presence of mannitol, as evidenced by the appearance of ice Ih reflections on the corresponding diffraction pattern (Figure 2a, diffraction pattern 2). A comparison of the powder diffraction pattern of samples of frozen THF–water solutions [49] with that of the frozen “mannitol–THF–water” sample suggested that ca. 20–30% of the THF did not form a clathrate hydrate.

**Table 1 pharmaceutics-14-00495-t001:** Results of the thermal analysis experiments, aiming to study the melting behavior of the THF hydrate in ternary (solute (glycine–mannitol–trehalose–Pz)–THF–water) systems. Glycine, mannitol, and trehalose aqueous solutions were used as reference samples.

Formulation	Crystalline Phases (PXRD)	Melting Temperature ± s.d., °C	Comments
THF–water (19.1 wt % THF)	THF hydrate	4.2 ± 0.1 (4.2 ± 0.1 [49]; 4.5 °C [44])	Congruent melting of the THF hydrate in binary system
Glycine–water (5 wt % glycine)	ice I*h* + *β*-glycine	−3.5 ± 0.1 (−3.5 ± 0.1 [69]; −3.6 °C [70]) −0.8 ± 0.2	Glycine–water eutectic ice I*h* melting (liquidus line)
TheIMannitol–water (5 wt % mannitol)	ice I*h* + *β*-mannitol	−1.5 ± 0.1 (−1.5 °C [71,72]) −0.5 ± 0.2	Mannitol–water eutectic ice I*h* melting (liquidus line)
Trehalose–water (5 wt % trehalose)	ice I*h*	N/A * −0.2 ± 0.1	--- ice I*h* melting (liquidus line)
Glycine (5 wt %)–THF–water (19.1 wt % THF)	THF hydrate + *β*-glycine	−3.9 °C (−3.9 °C [49])	Ternary peritectic
		+3.0 ± 0.1	THF hydrate melting in ternary system
Mannitol (5 wt %)—THF–water (19.1 wt % THF)	THF hydrate + + *β*-mannitol	−2.4 ± 0.2	Ternary peritectic
		+3.2 ± 0.2	THF hydrate melting in ternary system
Trehalose (5 wt %)—THF–water (19.1 wt % THF)	THF hydrate	N/A *	----
		+3.5 ± 0.2	THF hydrate melting in ternary system
Pz (1.5 wt %)– THF–water (19.1 wt % THF)	THF hydrate + Pz	−2.0 ± 0.2	Ternary peritectic
		+4.0 ± 0.2	THF hydrate melting in ternary system

* Trehalose forms an amorphous freeze-concentrate on cooling and remains amorphous during subsequent freeze-drying.

#### 3.3.2. Determination of the Pz Formulations’ Properties 

The two-component solid dispersions containing Pz and the selected carriers were prepared in ratios varying from 1:4 to 1:1. Visual inspection of each sample revealed that stable freeze-dried cakes (Figure 3) appeared to be in accordance with the “ideal cake”: the lack of any signs of skin on the cake surface and collapsed inclusions near the bottom of the vial [73]. According to SEM data, all formulations consisted of two types of particles: micrometer-sized “noodles” (Pz) and the fragments that are common for the internal structures of freeze-dried cakes of the selected carriers (Figure 3).

According to PXRD data, in the system containing glycine and Pz, Pz remained crystalline, and glycine precipitated as the metastable β-modification, similar to what was observed in the system with glycine only (Figure 4 diffraction patterns 1 and 2; Figure 5c). In the formulation containing Pz and mannitol, mannitol precipitated as the δ-modification (DMANTL12 [68]) with an admixture of the α-modification (DMANTL08 [53]), although the experiments began with mannitol in its β-modification (DMANTL09 [48]) (Figure 4, diffraction patterns 3 and 4; Figure 5b). This differed from the precipitation of mannitol as the mannitol hemihydrate in the system without Pz (Figure 2a and Figure 4). In the system containing Pz and trehalose, Pz remained crystalline, an amorphous freeze concentrate was formed on cooling, and trehalose remained amorphous during freeze-drying in all the formulations (Figure 4, diffraction patterns 5 and 6; Figure 5a).

#### 3.3.3. Determination of the Residual THF Level of the Pz Formulations 

The permitted concentration limit of THF in a pharmaceutical formulation may not exceed 700 ppm [38]. Tests of the residual THF level in the samples by GC–MS have shown that 73 ± 7, 4000 ± 200, and 93 ± 8 ppm were present in the formulations with mannitol, trehalose, and glycine as carriers, respectively. The exceedingly large THF levels observed in trehalose-containing samples can be related to the fact that the freeze-concentrate with trehalose was amorphous. This amorphous phase likely contained unfrozen water and THF that did not form a THF hydrate, and which could therefore not be removed on freeze-drying. Trehalose was correspondingly excluded from further consideration.

#### 3.3.4. Estimating the Dissolution of the Pz Formulations with Mannitol and Glycine 

The main purpose of preparing the formulations of poorly soluble Pz was to improve the dissolution dynamics, which is directly related to the bioavailability of a drug [74]. During the preliminary experiments aimed at determining the reconstitution times of Pz formulations (3 or 5 mL of saline per vial), when attempting to dissolve the formulation with glycine, a white-yellow precipitate was formed. (Appendix A). Analysis of the isolated precipitate showed that the ClO_4_^−^ group of Pz was chemically substituted for Cl^−^ as a result of the interaction of glycine with the perchlorate anion in solution (see details in the Appendix A). Therefore, glycine was also excluded from further consideration when aiming for formulations to be used in nebulizers with saline solutions. We note that glycine remains a suitable carrier when using water for injection, not saline solutions.

These experiments revealed a fast dissolution (less than 5 s) for Pz formulations with mannitol (1:4); Pz formulations with mannitol (1:1) and (1:2) demonstrated slower dissolution (more than 10 s) with the need for vigorous shaking. Therefore, mannitol became the only suitable excipient, yielding the final system of choice as “Pz–mannitol–THF–water” (320 mg Pz, 1280 mg mannitol in 40.0 g of the THF–water mixture (19.1 wt % of THF)).

### 3.4. Nebulizer Test Using Formulation of Pz with Mannitol (1:4) 

At this stage we aimed to investigate the following:(1)To test the stability of the prepared inhalation solution based on the formulation of Pz with mannitol (1:4) under an external action (vibrating membrane) during a time period typical for the total duration of the inhalation treatment (10–15 min);(2)To estimate the spraying rate of the solution (mL/min). This is important, since if the spraying speed of the solution is very low (less than 0.2 mL/min), the therapeutically needed dose of a drug will not be delivered to the lungs during a reasonable duration of an inhalation treatment;(3)To estimate the dose released during 10–15 min of an inhalation treatment.

Test experiments with a nebulizer showed that no precipitation of Pz occurred from a solution of the Pz–mannitol (1:4) formulation during the typical time of the inhalation treatment (10–15 min), even if the supersaturation level of the solution was twice as high as the equilibrium saturation concentration of Pz (Appendix A). Using HPLC, we have shown that the concentration of the Pz solution in a container did not change after the inhalation treatment was over. Spraying speed was 0.75 mL/min (saline solution) or 0.69 mL/min (water for injection). Based on the Pz content remaining in the solutions within the nebulizer reservoir after the end of the total inhalation procedure, we obtained the emitted dose (the amount of Pz leaving the mesh nebulizer reservoir as an aerosol) of our samples. For a sample of the Pz formulation dissolved in 10 mL of saline solution or in water for injection, the emitted doses were 30 and 28 mg, respectively, when measured over a 10 min total inhalation treatment time. This emitted dose could be increased to 40 mg by prolonging the total inhalation duration to 15 min.

These tests prove that the formulations meet the standards required for a formulation intended for administration via a mesh nebulizer.

## 4. Conclusions

The formulations of poorly soluble drugs that are suitable for injections or inhalations can be prepared as solid dispersions with APIs by freeze-drying water–organic solvent mixtures that contain both the API and a carrier [49]. The choice of the organic solvent, the carrier(s) and the freezing–heating protocol is not straightforward. For example, it is difficult to bring antibiotics used in tuberculosis treatment to the market as dry powder inhalation formulations, since these drugs need to be given in the hundreds of milligrams range (daily dose up to 1000–1500 mg). This quantity is beyond the technical capabilities even for capsule DPIs. Moreover, peroral and intravenous administration routes do not allow one to deliver a large dose to the lungs [18,19]. We have solved this problem for a new anti-tuberculosis drug, Pz, by preparing a formulation that can be delivered using a nebulizer for inhalations.

We illustrated in this case study the possibility of using a formation based on a water–organic clathrate hydrate that is formed on freezing. Both the water and organic solvent readily sublime during freeze-drying, leaving the API and carrier as fine dispersions that are suitable for injections or inhalations. The formulations were prepared by conventional freeze-drying in vials, making the proposed method suitable for industrial scaling. We have also explained the principles of selecting an organic co-solvent and a carrier. A similar strategy of selecting the organic solvent and the excipient can be applied to other compounds with low water solubility. It is important that the chemical composition of the API remains the same in the formulation and that only pharmaceutically acceptable components are used as excipients.

The medical effects of our reported composites cannot be easily compared with those of pure Pz, since no inhalable forms for nebulizers can be prepared from pure Pz. However, one could possibly compare them with other formulations based on dispersed functionalized particles (see [27,30]). With a reliable and scalable method of obtaining formulations suitable for nebulizers now demonstrated, we hope that such further medical tests will be performed and reported by other researchers. As mannitol is a common pharmaceutical excipient, we do not expect our proposed formulation to pose any series difficulty in the translation from bench to clinic. Moreover, we do expect a significant impact of this new formulation on the efficacy of tuberculosis treatment since it allows for an increase of the dose that is delivered directly to the target (lungs) whilst decreasing hepatotoxicity by avoiding digestive metabolism.

## Data Availability

The data presented in this study is contained within the article and Appendix A. The primary data presented in this study are available on request from the corresponding author Dr. Andrey Ogienko (andreyogienko@gmail.com).

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
