# Peer review of "Clathrate Hydrates of Organic Solvents as Auxiliary Intermediates in Pharmaceutical Research and Development: Improving Dissolution Behaviour of a New Anti-Tuberculosis Drug, Perchlozon"

_pharmaceutics, 2022, doi:10.3390/pharmaceutics14030495_

Round 1

Reviewer 1 Report

(1) In Figure 1a, many internal are not cited, such as figure 1a3 and figure 1a4. Please explain, and what is the difference between figure 1a1, 1a2 and figure 1a3 and 1a4?

(2) Figure 1a and figure 1b need to be explained in the text. Under ambient pressure, different temperature conditions show similar characteristic peak curves, but under 100 mTorr pressure, they show different characteristic peaks. Please explain in detail.

(3) The value of scale bar in Figure 2 should be directly marked on the figure, which is more conducive to reading and easier to be distinguished quickly.

(4) Why are the positions of the characteristic peaks of Fig. 4f and Fig. 4g almost similar? However, there are great differences between Fig. 4i and Fig. 4j, one is in crystalline form and the other is amorphous. Please explain and add in the text.

(5) The PXRD in Figure 1a and 1b has many key characteristic peaks. Please mark the location in the images and appropriately state and explain it in the text.

(6) In the XRD curve of Fig. 4, some important eigenvalue positions need to be displayed.

Author Response

please see file attached.

Reviewer 2 Report

  • A manuscript must include all data so that any reader can be able to reproduce the experiments. From my point of view, this does not happen in this manuscript.
  • The authors have designed the workflow in a poor way, also, it would be good for readers to have a graphical representation of the work.
  • Although the aim of this study was to investigate the therapeutic potential perchlorate as clathrate as anti-tuberculosis drug, there was no experiment conducted to prove this aim.
  • According to the main purpose of study, the authors should be focused on the evaluation of the in vitro and in vivo performance of the optimized formula.
  • The prepared powders were freeze-dried which produces normally low-density powder, authors  did not conduct the angle of repose study for the optimized powder in addition to the density which is more important to predict the flow ability during nebulization.
  • What statistical analysis that can be performed to see the effect of different formulation variables on product performance to get optimized formulation.
  • The formulation was delivered using nebulizer, so I guess the drug was in solution form before nebulization. I am not quite clear if the solution form of plain drug and prepared formula would behave. Needs to clarify it.
  • Results and discussion section is very poor and need much more explanation and details
  • Language and spelling editing have to be performed by authors to consider the manuscript for publication. English language is poor and need to be must be revised, and a lot of typos must be corrected
  • Authors have cited around 90% of references before 2020, including one reference at 1968. The newest one is 3 years ago. All references need to be updated

Author Response

Please see file attached.

Reviewer 3 Report

In this paper, the authors use the case study of a new anti-TB drug, Pz, to justify the selection of THF as a co- solvent and mannitol as a carrier that enabled us to prepare formulations of this poorly soluble compound that are suitable for injections or inhalations. The manuscript addresses an interesting question, but there are still some detailed problems to be rectified and revised.

Majors:

1.In line 63,the author mentioned that “When processing thermo-labile compounds, however, freeze-drying is likely the only usable technology.” This theory is not accurate. Spray drying is also suitable for thermo-labile compounds due to the short thermal contact time.

2.In Samples preparation part, the author mentioned Perchlozone was synthesized according to [13] and Glycine (analytical grade, JSC “Reakhim”) was recrystallized as described in [28]. You should briefly introduce the preparation method.

  1. In 2.2.1. Ultrafine Pz preparation method (TFF+FD), What do you mean with (TFF+FD)? The abbreviations TFF and FD do not appear in the before text.
  2. In 2.3. TA experiments part, what do you mean withTA? The abbreviations TA do not appear in the before text.

5.In line 153, the author mentioned that “The TA experiments were carried out using a home-made instrument described in [26].”In your description below, I do not quite understand the composition of this home-made instrument and suggest that the author draw a diagram showing the instrument or specify this instrument.

Minors:

  1. We note that the title of the article uses two sentences, we suggest changing it to one sentence.
  2. In the part of Materials and Methods, Please list materials and methods separately.
  3. In line 134, “Pz (300 mg) was dissolved by stirring at ca +30ºÐ¡ in 20.0 g of the THF/water mixture (19.1 wt % of THF, THF hydrate composition) in a transparent borosilicate glass vial (Sci/Spec, B75525)”, I don't quite understand the what the ca +30ºÐ¡ mean?
  4. In line 147, China)) an extra parenthesis is added here.
  5. In the part of 2.4. X-ray powder diffraction analysis, although the determination conditions were listed, the sample information was not listed. Please supplement the sample information.
  6. In Results and Discussion part, I suggest the author divide this part into different parts consistent with materials and methods part.
  7. In line 287, the author mentioned that “and later re-refined in”, I confused whether after re-refined in lacks some vocabulary?

Author Response

See file attached

Reviewer 4 Report

The manuscript pharmaceutics-1554211 is an original article focused on selecting solvents and excipients to prepare Pz formulations with improved dissolution behaviour, and appropriate for intravenous/respiratory (nebulizers) delivery. Results show that using THF as a co-solvent and mannitol as a carrier, appropriate Pz formulations for injections or inhalation were obtained by conventional freeze-drying in vials. Overall, this article is well organized, easy to read, and the topic fits within the scope of Pharmaceutics. The methods are adequately described; the results are clearly presented; the conclusions are supported by the results; and the references are appropriate. Therefore, I would recommend the publication of this work after addressing the following minor comments:  

  1. There are some typos such as: page 1, line 39, Mycobacterium tuberculosis should be written in italic; on page 2, line 91, there is a space between +4.5 and °C that is not necessary; on page 5, line 206, Pz should be used instead of perchlozone; and on page 6, line 245, In situ should be in italic.
  2. There is a more recent Global Tuberculosis report (https://www.who.int/teams/global-tuberculosis-programme/tb-reports), thus the data presented in the introduction (page 1, lines 40-41) should be updated
  3. Finally, in the conclusions, the authors should address the main challenges for the translation from bench to clinics, referring the impact of this formulation on tuberculosis treatment.

Author Response

see file attached

Round 2

Reviewer 1 Report

This paper can be accepted for publication.

Author Response

We thank for this evaluation. We tried to improve English. All modifications are shown directly in the uploaded revised version.

Reviewer 3 Report

The author has revised th manuscript. But there is still little issue to be improved as follows:

1.The author mentioned that “When processing thermo-labile compounds, however, freeze-drying is often the technology of choice, if spray drying is not applicable because of a polymorphic transformation / partial decomposition of a thermo-labile compound can occur even during a short thermal contact time. Our point was that even if thermal contact time is short, polymorphic transformations and/or partial degradation are possible.”  This point is not accurate, Wouldn't freeze-drying process affect the polymorphic transformation / partial decomposition of a thermo-labile compound?

2.In the part of Materials and Methods, you still don’t separate materials and methods.

Author Response

1. Wouldn't freeze-drying process affect the polymorphic transformation / partial decomposition of a thermo-labile compound?

As for partial decomposition of thermo-labile compounds, i.e. compounds unstable on heating, freeze-drying is more likely not to result in their decomposition.

As for polymorphism, yes, you are right, both techniques can affect the polymorphism, though differently. We tried to rephrase again.

2. We did not understand what exactly was requested, but tried to "separate materials from methods".

We tried to improve English.

All the changes are shown directly in the uploaded revised version.